# Ultra-high resolution imaging of thin films and single strands of polythiophene using atomic force microscopy

Vladimir V. Korolkov [1], Alex Summerfield [1], Alanna Murphy[2], David B. Amabilino[2], Kenji Watanabe [3], Takashi Taniguchi[3] & Peter H. Beton [1]

Real-space images of polymers with sub-molecular resolution could provide valuable insights into the relationship between morphology and functionality of polymer optoelectronic devices, but their acquisition is problematic due to perceived limitations in atomic force microscopy (AFM). We show that individual thiophene units and the lattice of semicrystalline spin-coated films of polythiophenes (PTs) may be resolved using AFM under ambient conditions through the low-amplitude ($\leq 1$ nm) excitation of higher eigenmodes of a cantilever. PT strands are adsorbed on hexagonal boron nitride near-parallel to the surface in islands with lateral dimensions ~10 nm. On the surface of a spin-coated PT thin film, in which the thiophene groups are perpendicular to the interface, we resolve terminal $CH_3$-groups in a square arrangement with a lattice constant 0.55 nm from which we can identify abrupt boundaries and also regions with more slowly varying disorder, which allow comparison with proposed models of PT domains.

[1] School of Physics and Astronomy, University of Nottingham, Nottingham NG7 2RD, UK. [2] School of Chemistry, University of Nottingham, Nottingham NG7 2RD, UK. [3] National Institute for Materials Science, 1-1 Namiki, Tsukuba, Ibaraki 305-0044, Japan. Correspondence and requests for materials should be addressed to V.V.K. (email: korolkov.vladimir@gmail.com) or to P.H.B. (email: peter.beton@nottingham.ac.uk)

Polymers, both synthetic and natural, are ubiquitous materials whose properties are strongly influenced by packing, conformation, and monomer composition of individual macromolecules. The ability to acquire real-space images of the microstructure of these materials with molecular-scale resolution is required to advance the understanding and control of their local ordering, a key element in the precise engineering of polymer properties[1–3]. These considerations are particularly important for conducting and semi-conducting organic polymers[4–8], which are used widely in organic photovoltaics[9,10], light-emitting diodes[11,12] and other flexible optoelectronic devices[13–16]. The polymer microstructure plays a crucial role in the transport of charge carriers and excitons, thus determining device performance[17]. The close relationship between microstructure and carrier mobility is well established through X-ray diffraction studies of, e.g., polythiophene (PT) and its derivatives, which show that domains with typical dimensions of a few tens of nanometres are formed[5,18]. However, the nature of the boundaries between domains, which are crucial to an understanding of carrier transport at a macroscopic level[17], cannot be determined from reciprocal space methods. In principle, scanning probe microscopy (SPM)[2,19–21] is ideally suited to such structural studies, but the resolution which is routinely available using conventional atomic force microscopy (AFM) under ambient conditions is a significant constraint to the acquisition of images, which reveal ordering of polymers at the molecular and sub-molecular scale.

In many examples of the high-resolution structural characterization of artificial polymers using SPM, scanning tunnelling microscopy (STM) is the preferred imaging modality, as, under a range of operating conditions including liquid[22] and UHV[23–26], images revealing single polymer strands may be acquired. This body of work also includes several studies of polymers formed by on-surface synthesis on noble metal surfaces[27–30]. AFM has also been used to acquire images of single strands of synthetic polymers[1,2,31–37], whereas the introduction of torsional tapping-mode AFM imaging[38] led to the acquisition of images of single molecules and semicrystalline domains of polyethylene[3,39] achieving resolution down to 0.37 nm in air. This approach, however, requires specialized T-shaped ultrasharp cantilevers with carbon-whisker tips and a modified AFM with improved noise characteristics of the lateral deflection signal[39]. Previously, we have shown that higher eigenmodes can provide a route to higher resolution on molecular films and two-dimensional (2D) materials under ambient conditions[40,41] and, in addition, Proksch and colleagues[31] have recently shown that bimodal tapping imaging, where two eigenmodes are excited simultaneously, can provide high spatial resolution of semicrystalline polypropylene and polyethylene alongside their mechanical properties. A comprehensive overview of significant advances in polymer research achieved with AFM has been published by Wang and Russell[1].

In this study we show that images with ultra-high resolution can be achieved in a conventional tapping mode (AC mode) using standard silicon nitride probes in conjunction with a commercially available instrument by exciting higher-order resonances of the cantilever; the resulting images provide new insights into the molecular-scale structure and order of semiconducting polymers. The polymers chosen for our study are two commercial PTs, materials that are widely used to fabricate organic solar cells and field effect transistors, and we demonstrate this level of resolution for both single polymer chains deposited on the atomically flat surface of hexagonal boron nitride (hBN) and at the surface of a spin-coated thin film with a roughness of a few nanometres.

## Results

**Polythiophene on hexagonal boron nitride.** PTs are formed from thiophene units that are covalently linked at the 2- and 5-positions of the heterocycle (see structural diagrams in Fig. 1). The polymer backbone is often functionalized with alkyl chains at the 3-position of thiophene ring for improved solubility and processability. Regio-random poly(3-decylthiophene-2,5-diyl) (P3DT) was purchased from American Dye Source, Inc., and deposited on hBN flakes from solution; details of sample preparation are provided in Methods. Tapping-mode AFM imaging was performed under ambient conditions immediately after sample preparation using an Asylum Cypher S AFM (Oxford Instruments-Asylum Research, Santa Barbara, USA). To achieve ultra-high-resolution images we use Multi75Al-G cantilevers (Budget Sensors, Bulgaria) driven at either the second or third eigenmode, corresponding to resonant frequencies of ~450 kHz and ~1.3 MHz, respectively, and oscillation amplitudes less than ~1 nm. We have also used high-frequency Arrow UHF probes (NanoWorld AG, Neuchâtel, Switzerland) at their first eigenmode of ~1.4 MHz. The amplitude sensitivity for the first (65 nm/V Multi75Al-G and 10 nm/V Arrow UHF) and third eigenmode (17 nm/V Multi75Al-G) was estimated using the thermal method as implemented in GetReal calibration procedure (Oxford Instruments-Asylum Research). The exact scanning parameters were optimized for each AFM scan and are provided in figure captions.

A typical overview AFM scan of PT molecules is presented in Fig. 1a. An inset shows a 5 nm × 5 nm AFM tapping-mode scan of the underlying hBN surface acquired from the same area. The lattice vectors of the hBN lattice are overlaid on Fig. 1a (see also schematic in Fig. 1d). AFM reveals domains with parallel bright lines, which we attribute to P3DT polymer strands. Domains have typical dimension of 10–30 nm, an inter-strand separation of $1.95 \pm 0.02$ nm and an apparent height of $0.33 \pm 0.02$ nm (Fig. 1c) consistent with the typical thickness of a flat-lying aromatic molecules[40,42]. The decyl side chains appear as low-contrast features, which are more difficult to resolve, but may be distinguished as faint lines running perpendicular to the bright P3DT backbones (see Fig. 1b, a digitally cropped image from Fig. 1a). We have observed that such contrast behaviour is typical for cantilevers operating at the first eigenmode and relatively large oscillation amplitudes of 15–20 nm (250–300 mV for Multi75Al-G probes) or 7–9 nm (700–900 mV for Arrow UHF probes).

From a comparison with the orientation of the hBN lattice shown in Fig. 1a inset, it is clear that P3DT strands are oriented perpendicular to the hBN lattice vectors in one of three orientational domains. This implies, given the structure of an individual PT molecule (Fig. 1d), that the decyl side chains lie parallel to hBN lattice vectors similar to the arrangement observed for *n*-alkanes on hBN[43]. This suggests that the placement of P3DT molecules and, hence of the PT backbone itself, is determined, at least in part, by the preferred orientation of the decyl side chains. As is evident from the AFM scan (Fig. 1a), there are many P3DT molecules that span, and thus connect, two or more domains, in some cases with a switch in orientation. Hairpin bends with a high bending curvature are also observed, similar to those reported in previous STM studies of PTs on graphite[22,44].

We have further imaged individual P3DT molecules using a combination of small oscillation amplitude (typically 0.3–0.5 nm (20–30 mV)) and the third eigenmode of the Multi75Al-G cantilevers to reveal the intramolecular structure of the PT backbone (see Fig. 1f). We have found that this combination produces images that are significantly different from those acquired with the first eigenmode as shown previously (Fig. 1a).

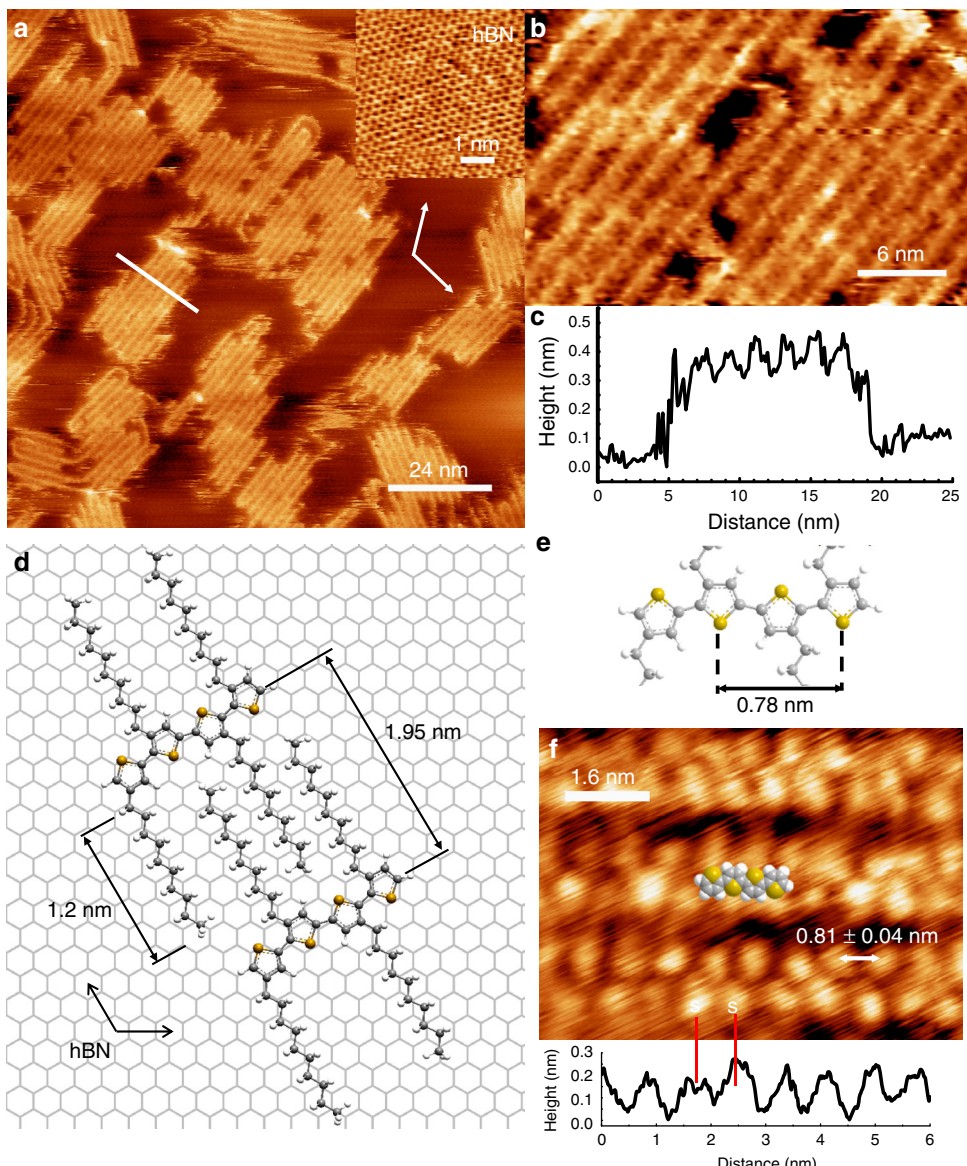

**Fig. 1** High-resolution AFM images of P3DT adsorbed on the surface of hBN. **a** An overview height scan of P3DT assembled on hBN, scan rate 6.51 Hz, 1024 × 1024 px; inset shows lattice frequency shift image of hBN acquired in FM-AFM tapping mode, scan rate 39 Hz, 512 × 512 px; both images were acquired with the same Arrow UHF probe oscillating at fundamental frequency of 1.42 MHz. **b** A selected area from scan **a** showing individual PT molecules and low-contrast features associated with decyl side chains. **c** Profile along the line highlighted in **a** showing molecule-to-molecule separation of 1.95 nm. **d**, **e** Structural models for P3DT assembly on hBN with corresponding dimensions calculated using the AM1 semi-empirical method. **f** Ultra-high-resolution AFM height scan of three individual P3DT molecules. Image (**f**) was acquired using third eigenmode of Multi75Al-G probe oscillating at 1.255 MHz and a setpoint of 24 mV. Scan rate 19.5 Hz, 1024 × 1024 px

As opposed to the latter, these images show that a single P3DT strand is composed of two rows of bright alternating features with a period of 0.81 ± 0.04 nm. When compared with a structural model of a PT backbone (Fig. 1e), it is concluded that the centres of these features are likely associated with sulphur atoms. Indeed, the S–S interatomic distance is in good agreement with the value, 0.78 nm, measured for terthiophene[45,46], 3,4',4"-trimethyl-2,2':5',2"-terthiophene[47] and septithiophene[48]. The strong contrast due to sulphur atoms is likely due to their larger size (atomic radius of 100 pm vs. 70 pm for carbon) and the lone pair of electrons localized on the that atom. A model structure of quaterthiophene is overlaid on Fig. 1f, to highlight positions of sulphur atoms that are marked for clarity on both the image and the profile. In summary, we argue that the contrast originates interactions with the occupied

3p orbitals of sulphur atoms producing an exaggerated width for the PT backbone.

To strengthen the assignment of the high-contrast features in Fig. 1f to S atoms we have performed density functional theory calculations, which include the Grimme[49] dispersion correction (B3LYP-D, 4–31G-1D basis set; see Supplementary Methods for more details) for both bithiophene and terthiophene molecules adsorbed on a finite hBN slab with H-terminated edges (Fig. 2a— data for terthiophene are shown; see Supplementary Fig. 1 for bithiophene). All calculations were performed with Firefly QC package, which is partially based on the GAMESS (US) source code[50,51]. The relaxed structure of terthiophene on hBN exhibit a *trans*-gauche conformation with a torsional angle of 173° (Fig. 2b) and sulphur atoms located farther away from the surface when compared with the carbon atoms. There is no published data on

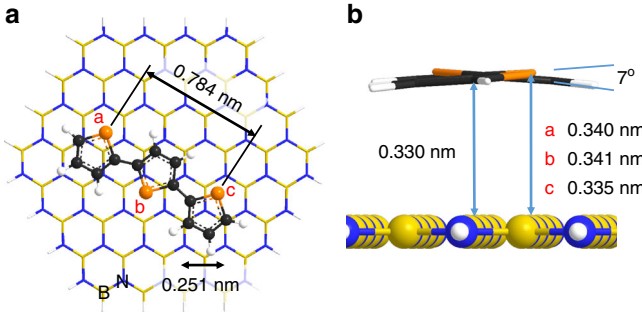

**Fig. 2** Calculated structures of *trans*-gauche conformation of terthiophene adsorbed on a hBN slab. **a** Top view and **b** side view along the molecular axis showing rotational distortion of the molecule. Both the calculated torsional angle between thiophene units and the molecule–substrate distances are shown

oligothiophene geometries on hBN or similar surfaces, but previous spectroscopic studies of quaterthiophene suggest a *trans*-planar conformation[52] on the Ag(111) surface. However, in its crystalline form, terthiophene exhibits a twisted *trans*-geometry with a torsional angle of about 171°–174°[46]. Bithiophene is known[53] to have a completely flat (torsional angle of 0°) structure in the solid state and a twisted structure in the gas phase with a torsional angle of 148° ± 3°. Our calculations predict that bithiophene should have a twisted trans-gauche geometry when adsorbed on hBN with the same torsional angle of 173° calculated for terthiophene. Therefore, our calculations predict a twisted geometry with protruding sulphur atoms, thus accounting for the observed contrast in AFM scans of P3DT on hBN. In addition, the calculated S–S separation (0.78 nm) and molecule–substrate spacing (0.33 nm) are in good agreement with our measured values.

We also show in Fig. 1d a calculated model of interacting P3DT molecules based on two terthiophene molecular blocks with decyl side chains, which was optimized using the semi-empirical method AM1 as implemented in Firefly QC package. The calculated separation of neighbouring molecules is in excellent agreement with the observed value of 1.95 ± 0.02 nm.

**Resolution of molecular order in PT spin-coated thin films**. We have also investigated the surface of a spin-coated thin film of regioregular (>95%) poly(3-hexylthiophene-2,5-diyl) (P3HT). For these investigations, the P3HT film was deposited on a mica substrate on which a PEDOT:PSS layer had been previously deposited by spin coating (see Methods for more details). P3HT films formed in this way are expected to have similar morphological properties as the active layers in photovoltaic devices. Figure 3 shows AFM images of the surface of the resulting thin film. Topographic images (Fig. 3b) reveal a roughness of a few nanometres and topographically bright islands with typical dimensions of ~10–20 nm. In general, the roughness presents a challenge to the acquisition of high-resolution images in tapping-mode AFM. However, as shown Fig. 3a we are able to acquire images in the phase channel in which sub-molecular features are resolved. Here we scan in AC mode regulating the height of the cantilever to maintain a constant amplitude of oscillation, while also acquiring the variation of the phase of the oscillating cantilever relative to the drive signal. Phase images are conventionally interpreted as providing contrast between different material properties that modify, locally, the damping of the oscillating cantilever[54]. In such images, we clearly resolve a square lattice with approximate dimension of 0.55 nm × 0.55 nm. These regions

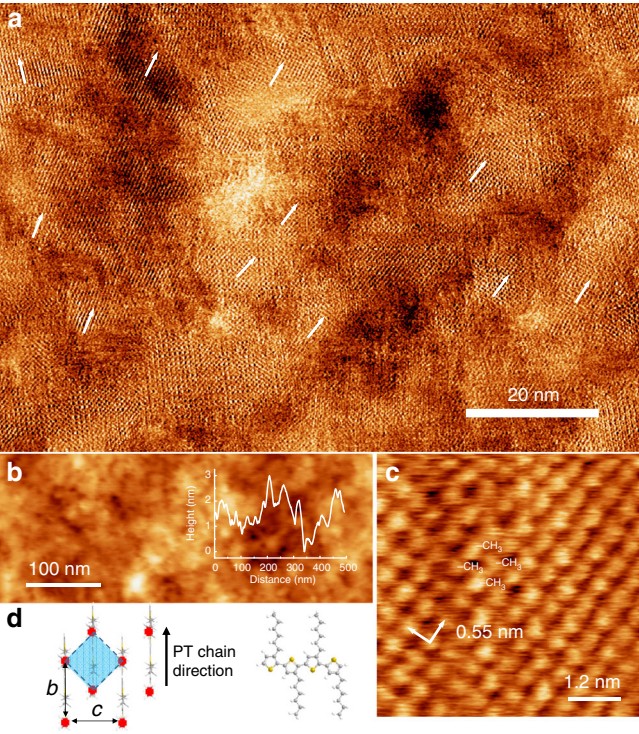

**Fig. 3** AFM images of the surface of a semi-crystalline spin-coated film of P3HT. **a** High-resolution AFM scan (phase image) showing differently oriented domains with square symmetry; image acquired with Multi75Al-G probe oscillating at the third eigenmode of 1.255 MHz and setpoint of 9 mV; scan rate 4.34 Hz, 1536 × 1536 px. **b** Large-scale AFM height image showing overall topography and roughness of the sample scanned with Multi75Al-G probe oscillating at fundamental frequency of 67.7 kHz and setpoint of 280 mV; scan rate 1.5 Hz, 512 × 512 px. **c** High-resolution AFM height image showing terminal methyl groups of hexyl chains attached to the P3HT backbone; image acquired with Multi75Al-G probe oscillating at the third eigenmode of 1.255 MHz and setpoint of 10 mV; scan rate 1.5 Hz, 512 × 512 px. **d** Schematic showing structural model for P3HT organization at the molecular level; the dimensions $b$ and $c$ are the lattice constants as determined in previous diffraction studies ($b \approx c = 0.78$ nm)[18]; the shaded region is the unit cell of the square lattice resolved at the surface. The expected period $= b/\sqrt{2} = 0.55$ nm is in agreement with our measurements

can be resolved in large area images (Fig. 3a) where we see several domains with different orientations of the square lattice.

It is has been established[5,18,55–58] from diffraction and spectroscopic studies that regioregular P3HT thin films form semi-crystalline domains in which the plane of the PT backbone is perpendicular to the substrate, forming an in-plane π–π stacked organization. In this arrangement, neighbouring sulphur atoms in a P3HT chain point alternatively up (away from the surface) and down (towards the surface). Neighbouring π–π stacked P3HT chains are arranged so that an 'up' sulphur atom is aligned with a 'down' sulphur in the adjacent chain. This leads to a crystal structure with an in-plane square lattice with dimensions determined from diffraction measurements to be $b = c = 0.78$ nm (see Fig. 3d) and two P3HT chains per unit cell[18]. We attribute the bright features resolved in Fig. 3a to the terminal CH$_3$-groups of the hexyl sidechains, which are attached to the P3HT backbone; these form a regular arrangement with an alkyl chain attached to thiophene groups at the vertices and centre of the surface unit cell (see Fig. 3d; the terminal points are marked as red circles). Accordingly, these terminal methyl groups form a square array with an expected lattice constant of $b/\sqrt{2} = 0.55$ nm

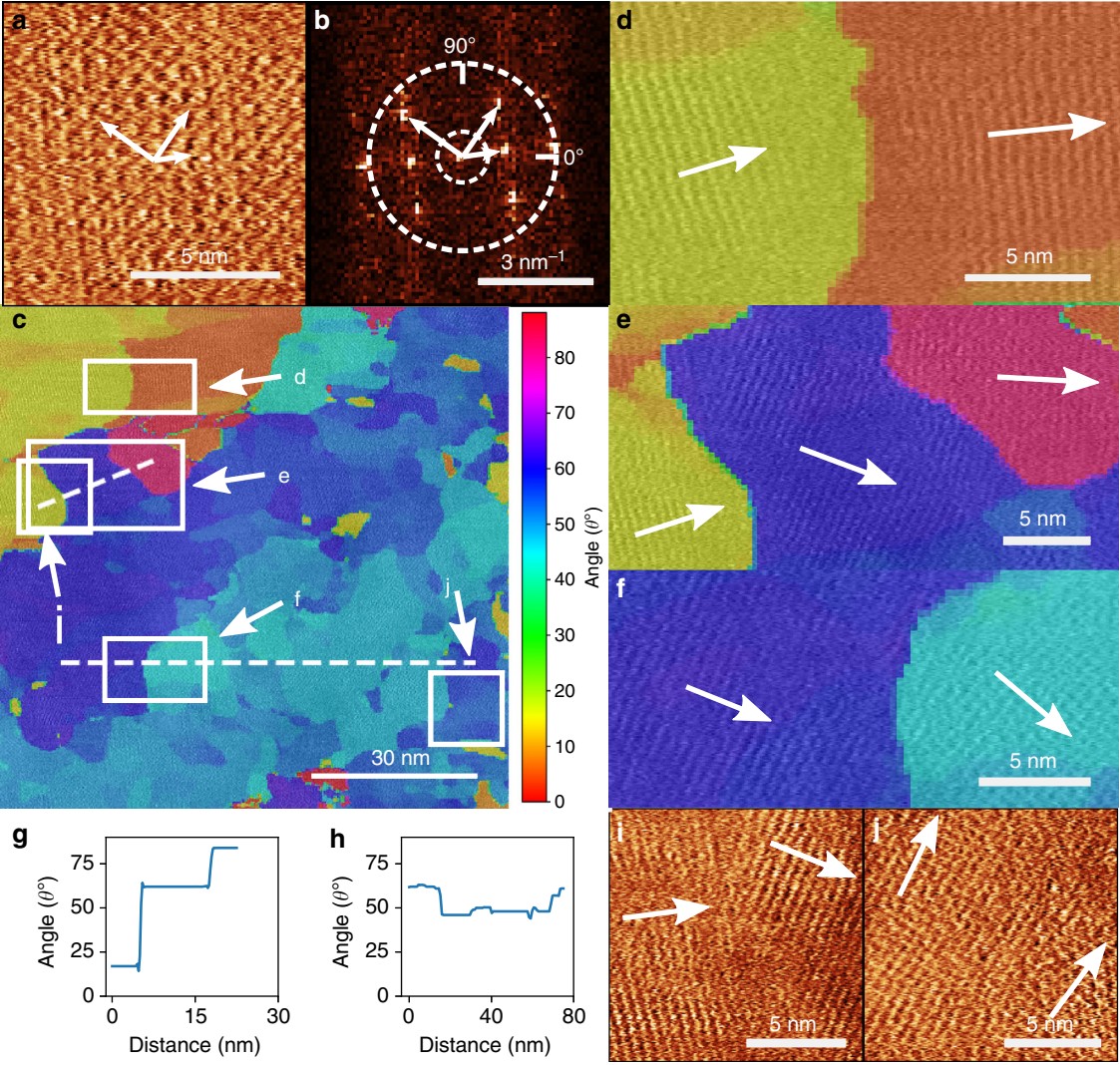

**Fig. 4** Fourier processing of P3HT images. **a** Tapping-mode phase-channel AFM image of P3HT, the white arrows indicate the angles of the Fourier peaks shown in **b**. **b** 2DFFT power spectrum of image (**a**) showing peaks in reciprocal space used to calculate domain orientation. A given spatial frequency range is used to isolate peaks of interest (e.g., as shown by the annulus between the white dashed circles). **c** Colour map of P3HT domain angular orientation extracted by performing a sliding 2DFFT across Fig. 3a with a kernel step size of 5 pixels using an 8.1 × 8.1 nm (124 × 124 pixel) zoomed image and plotting the angle of the most intense peak within the range $\theta = 0$–$90°$ (see colour scale). **d–f** Cropped regions as indicated in image (**c**) showing the domain structure at domain boundary regions with a transparent colour map overlaid, indicating the computed domain angle (colour-angle scale is the same as in **c**). **g** Interpolated line profile extracted from the upper dashed line in **c** showing the large angular change at the boundaries between the three P3HT domains shown in **e**. **h** Line profile along the lower dashed line in **c** showing small angular variation along the bottom region of the image. **i** Real-space tapping-mode phase-channel images showing an abrupt change in angle between two P3HT domains at the labelled region in image **c**. **j** Real-space tapping-mode phase-channel images showing a small angular change between a continuous P3HT domain at the position indicated by the white box in **c**

(as highlighted in Fig. 3d) in excellent agreement with the spacing determined from our AFM images (see Fig. 3c).

Real-space images of the molecular order within a PT semi-crystalline domain provide insights into the local order (and disorder) within the film. In Fig. 3a, there are many areas where the orientation of the square lattice may be clearly resolved, typically in domains with lateral dimensions of 10–20 nm. The local orientation of each domain is highlighted by an arrow in Fig. 3a and we see that across much of the image there is a relatively small angle between the orientation of neighbouring domains. This suggests that the order in many neighbouring domains is correlated and we also find that in the boundary regions between these domains the orientational order varies in a quasi-continuous manner (see Discussion below). However, in some areas, e.g., the top left of Fig. 3a, there is a boundary where the orientation appears to change more abruptly.

To analyse further the variation in domain order, a 2DFFT (2D fast Fourier transform) analysis of the P3HT image in Fig. 3a was performed using a similar method to our previous analysis of graphene/hBN heterostructures[59]. The high-resolution AC-mode phase-channel image in Fig. 3a was analysed by extracting a regular array of 8.1 × 8.1 nm (124 × 124 pixel) zooms of the main image (referred to as kernels); the centre of each kernel is then shifted by 5 pixels in both the $x$ and $y$ directions across the entire image and a 2DFFT is calculated at each position (see Fig. 4). A typical kernel and its associated 2D spectrum are shown in Fig. 4a and b, respectively. The 2DFFT (Fig. 4b) shows intense spots with a square symmetry; the spots at the vertices of the square correspond to a real-space periodicity close to the expected 0.55 nm. A peak detection algorithm was then used to extract the orientation and wave vector of the peaks of each 2DFFT, which are associated with the periodicity of the P3HT lattice (the search

is conducted within the annulus defined by the dotted lines in Fig. 4b; see Methods for more details of the algorithm), allowing a systematic determination of the local orientation of the semi-crystalline lattice across the image. Figure 4c shows a colour-scale map of the domain rotation angle overlaid on the image (Fig. 3a), revealing the angular distribution of the P3HT domains. This map confirms the qualitative impression from Fig. 3a that over much of the image the orientation of the lattice varies slowly; these areas correspond to those in various shades of blue in the lower and right of Fig. 4c. In addition, line profiles of the domain angle can be extracted (e.g., along the horizontal dotted line in Fig. 4c) and show (Fig. 4h) that over this region the domain orientation, while not uniform, fluctuates only by angles within a range of ~10°. In contrast, in the region in the top left there is an abrupt jump in orientation, which is apparent in both the colour map and the profile (Fig. 4g).

To highlight further the differences between the boundaries in the upper left of Fig. 4c and those between domains within the blue region, which have small variations in angle, we show zoomed areas of the colour map (see Fig. 4d, e and f; these are highlights of the areas identified in Fig. 4c) and also zooms of the phase-channel images (Fig. 4i, j). At the boundary resolved in Fig. 4i we see the lattices of the two domains terminating at an abrupt boundary; the change occurs over a length scale comparable with the lattice dimensions. This is the type of domain boundary envisaged in many models[17] of disorder in semi-crystalline semiconductor polymers and represents a simple way in which orientational order is randomized; neighbouring domains have clearly distinct order and well-defined boundaries. In contrast, the boundary region in Fig. 4j shows no abrupt changes, although it is clear in this image that the lattice in the bottom right and top left of the image differ (by ~5°). In fact, the lattices corresponding to the two different orientations merge in a quasi-continuous manner, with a slow variation of the lattice direction over a length scale of order 5 nm. It is noteworthy that the lattice points may be resolved throughout this extended boundary region, and in many regions it is clear that the local square order and co-ordination is retained even where the orientation is varying.

The slowly varying lattice vectors in Fig. 4j suggest that rather than treating these regions as distinct domains, they may be more appropriately described as a single 'super-domain' with a high density of crystalline defects, which lead to a splaying of the local orientational order. In fact, all of the boundaries within the large blue region in Fig. 4c behave similar to those in Fig. 4j, suggesting that there is orientational order over the dimensions of this area (> 100 nm), but that this order fluctuates on a length scale of ~10 nm and with an angular peak-to-peak amplitude ~10°. These images provide an alternative picture of disorder to the commonly accepted basis for modelling (see e.g., ref. [17]); although we see domains with local order that is preserved only over ~10 nm, in agreement with diffraction measurements[5], the orientation of neighbouring 10 nm-scale domains is correlated within a super domain with much larger dimensions.

## Discussion
Overall, we have demonstrated that the use of higher eigenmodes in tapping-mode ambient AFM can be successfully employed to characterize both individual polymer strands down to a single-atom level and also the ordering of a semi-crystalline polymer with technological relevance. The combination of AFM and solution deposition provides a simple and high-resolution approach to characterizing the structure of polymers. This approach is complementary to the acquisition of images using a combination of electrospray deposition and SPM under ultra-high vacuum conditions[24,26], whereas the compatibility of our method with solution deposition and technologically relevant

spin-coated thin films offers some potential advantages. The real-space images are in good agreement with previous studies[5] but provide new insights into the nature of the boundaries between domains that are not easily determined using reciprocal space methods. Our results highlight the important role for high-resolution AFM in determining the properties of polymer strands and thin films of technological relevance, and we anticipate future progress in correlating device performance with structural properties at the sub-molecular scale based on this technique.

## Methods
**P3DT deposition on hBN**. The hBN flakes were transferred by exfoliation onto a silicon wafer terminated by a $SiO_2$ layer with thickness of 300 nm. Following exfoliation, the substrate was cleaned by brief flame annealing with a butane torch. The polymer was deposited from a toluene solution (2.5 µg/ml); deposition times ranged between 10 and 30 s (a time of 10 s results in a fractional coverage of 30–40% of hBN).

**P3HT spin-coated films**. P3HT (regioregular > 95%) was purchased from Sigma Aldrich and used without further purification; PEDOT:PSS (HTL Solar) was purchased from Ossila. Spin-cast devices were prepared using a POLOS spin150i spin coater. P3HT (23 mg ml$^{-1}$) was dissolved in chlorobenzene and stirred overnight in darkness at 60 °C. PEDOT:PSS (1.0–1.2 wt.%, aqueous solution) was filtered through a polyvinylidene difluoride filter (0.45 µm pore size) and deposited onto mica substrates by spin casting (40 µl, 3000 r.p.m., 30 s, additional 30 s to dry). The films were annealed on a hot plate for 10 min at 130 °C. P3HT was deposited by spin casting (60 µl, 550 r.p.m., 30 s, additional time for drying as needed, 3000 r. p.m. for 30 s). The samples were imaged before and after annealing. Annealed films were heated on a hot plate for 15 min at 130 °C; we observed no significant differences between annealed and non-annealed films.

**Extraction of Fourier peaks**. Following the sliding 2DFFT operation on each kernel position (see main text), the power spectrum at each position was rotationally interpolated at 1° increments and only spatial periodicities in the range 0.4–1.5 nm were retained to remove periodic noise due to scan artefacts and high-frequency noise along the raster direction (i.e., at 0°). Maximum intensity values were then extracted as a function of rotation angle along with the spatial frequency at the position of the most intense peak. For peaks with a spatial period in the range 0.4–0.6 nm (i.e., the $b/\sqrt{2} = 0.55$ nm lattice constant of the square P3HT lattice), a modulo 90° was then applied to the peak angle due to the square symmetry of the lattice. If the strongest intensity peak fell outside of this periodicity range, then 45° was also subtracted from the angle value before the modulo 90° operation, as this is associated with spatial frequency components along the $b = c = 0.78$ nm direction of the P3HT lattice. The angles of the strongest Fourier peaks were then plotted at the position of the kernel centres using a cyclic colour scale in the range 0–90°. Raw AFM data were extracted from.ibw files using Gwyddion[60], and further Fourier processing and data presentation was performed using a Python script and presented using Matplotlib[61].

## Data availability
The raw data for the AFM images may be accessed through the University of Nottingham Research Data Management Repository at https://doi.org/10.17639/nott.6183.

## Code availability
The code used to analyse the data may be accessed through the University of Nottingham Research Data Management Repository at https://doi.org/10.17639/nott.6183.

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

## Acknowledgements

This work was supported by the Engineering and Physical Sciences Research Council [grant numbers EP/N033906/1 and EP/P019080/1] and the Leverhulme Trust [grant number RPG-2016-104]. K.W. and T.T. acknowledge support from the Elemental Strategy Initiative conducted by the MEXT, Japan, and the CREST (JPMJCR15F3), JST.

## Author Contributions

V.V.K. and P.H.B. conceived the experimental project with further contributions from D.A. and A.M. The AFM imaging and supporting calculations were carried out by V.V.K. The analysis of domain structure was performed by A.S. A.M. and D.A. designed and prepared the spin-coated samples, and T.T. and K.W. grew the hBN crystals. V.V.K., A.S. and P.H.B. analysed the data and wrote the paper, with revisions and comments from all authors.

## Additional information

**Competing interests:** The authors declare no competing interests.

**Journal Peer Review Information:** *Nature Communications* thanks Nic Mullin, Roger Proksch, and other anonymous reviewer(s) for their contribution to the peer review of this work. Peer reviewer reports are available.

