## [Peer Review File · Nature Communications]

Reviewers' comments:

Reviewer #1 (Remarks to the Author):

Korolokov et al. have submitted a real space polymer resolution tour de force. This work presents some very surprising results that will spur significant follow-on work in a wide range of investigators interested in polymer structure and function. As such, it goes well beyond the scientific results for the commercially important thiophene the authors measured (this was already sufficiently exciting for Nature) - I believe it will rapidly be repeated for a large range of other polymer systems. As such, I think this work is very appropriate for Nature Communications.

The paper was well-written and easy to understand.

One small note - Please include the python scripts in your supplemental materials or better yet a Jupyter notebook. This will enable reproduction of the analysis for a much wider range of people.

Reviewer #2 (Remarks to the Author):

The authors present gorgeous high resolution images of semiconducting polymers obtained by exciting higher order resonances of the cantilever in otherwise conventional AFM. While these images are brilliant, the technique was previously published (Nature Chem and Nature Comm both in 2017) by this author group. As a result, this manuscript's potential impact relies entirely on the importance of using this imaging technique on semiconducting polymers. In this case, the authors image the polymers after deposition on hexagonal boron nitride. As has been previously demonstrated by many groups, the substrate influences the polymer order by epitaxy and as a result the structure they see is a combination of the polymer's natural tendency to crystallize and the epitaxial effect (and the lattice mismatch created by the substrate). Since the application of the polymers is on thin films of NOT boron nitride for transistors and in much thicker films (100nm) for optoelectronic applications, the utility of understanding how polymers crystalline onto boron nitride is of limited utility and not of sufficient impact for publication in Nature Communications.

Reviewer #3 (Remarks to the Author):

The manuscript describes the use of dynamic AFM, working at higher eigenmodes of the cantilever, to image the molecular and sub-molecular organisation of polythiophenes in two- and three-dimensional films. Polythiophenes are of broad interest due to their extensive use in organic electronic devices and have been the target for several in-depth structural investigations with a variety of techniques. While scanning probe methods have been previously used to image molecular organisation in 2D films of these materials, this is (to my knowledge) the first report of molecular resolution in real space on a 3D polythiophene film. This is significant, as (as the authors point out) it paves the way for correlating structure with device performance in the future.

There are some areas where further discussion would be useful and interesting, specifically:

1) In figure 1a, the apparent length of the polymer chains is quite variable – from <10nm to >100nm. Is this in line with the molecular weight and polydispersity of the material used?

2) The features in figure 1b appear quite markedly different to those in figure 1f (no alternating

features along the backbone in 1b, and no visible side chains in 1f). Is this difference due to the different imaging conditions (amplitude, eigenmode etc.) used for the different images? If so, some brief discussion of how the imaging conditions affect the features that are resolved should be included.

3) In the discussion of figure 1f, the rows of bright alternating features are assigned to sulphur atoms. While the spacing along the chain supports this interpretation, the spacing across the chain appears to be significantly larger in figure 1f than is suggested by the structural models in figures 1d and 2a. Some discussion of the possible origin of this discrepancy should be included.

It would also be useful to include some further information, specifically:

a) Colour scale values should be included in the figure captions for all AFM images, and a scale and units should be added to the section in figure 1f.

b) A section covering AFM should be included in the Methods, including typical parameters such as typical values of the amplitude setpoint, scan rate, scan size, number of pixels etc and details of how the amplitude sensitivity was determined for higher eigenmodes.

Overall, the data are of extremely high quality, the analysis is robust and well described and the manuscript is well written. I believe this paper will be of interest in the field, both in terms of the imaging methods employed and the structural information that is revealed.

Reviewer #1

Korolokov et al. have submitted a real space polymer resolution tour de force. This work presents some very surprising results that will spur significant follow-on work in a wide range of investigators interested in polymer structure and function. As such, it goes well beyond the scientific results for the commercially important thiophene the authors measured (this was already sufficiently exciting for Nature) - I believe it will rapidly be repeated for a large range of other polymer systems. As such, I think this work is very appropriate for Nature Communications.

The paper was well-written and easy to understand.

One small note - Please include the python scripts in your supplemental materials or better yet a Jupyter notebook. This will enable reproduction of the analysis for a much wider range of people.

We thank the referee for their supportive comments. We are very happy to provide to the Python scripts together with simple instructions for wider use. Our preferred route is to upload this to directory in a publicly accessible data repository (which will be linked to the paper and also contain the relevant raw images – see statement under Data Availability) rather than the Supplementary Information, but we are flexible on this point and happy to be guided by the referee and editor.

Reviewer #2 (Remarks to the Author):

The authors present gorgeous high resolution images of semiconducting polymers obtained by exciting higher order resonances of the cantilever in otherwise conventional AFM. While these images are brilliant, the technique was previously published (Nature Chem and Nature Comm both in 2017) by this author group. As a result, this manuscript's potential impact relies entirely on the importance of using this imaging technique on semiconducting polymers. In this case, the authors image the polymers after deposition on hexagonal boron nitride. As has been previously demonstrated by many groups, the substrate influences the polymer order by epitaxy and as a result the structure they see is a combination of the polymer's natural tendency to crystallize and the epitaxial effect (and the lattice mismatch created by the substrate). Since the application of the polymers is on thin films of NOT boron nitride for transistors and in much thicker films (100nm) for optoelectronic applications, the utility of understanding how polymers crystalline onto boron nitride is of limited utility and not of sufficient impact for publication in Nature Communications.

We are pleased that the referee recognises the high quality of our images. However, we were surprised that the referee had overlooked Figures 3 and 4 in which present images of polymer thin films in the spin-coated form which the referee correctly identifies as being of relevance to transistors and optoelectronic applications. The associated discussion starts at the top of p6 with the statement,

'We have also investigated the surface of a spin-coated thin film of regioregular (> 95%) poly(3-hexylthiophene-2,5-diyl) (P3HT). For these investigations the P3HT film was deposited on a mica substrate on which a PEDOT:PSS layer had been previously deposited by spin coating (see Methods for more details). P3HT films formed in this way are expected to have similar morphological properties as the active layers in photovoltaic devices.'

and continues for a further three pages. A particular highlight of our work is the demonstration that the lattice structure of P3HT thin films can be resolved using real-space imaging – we believe for the first time – and that this information can be used to identify the nature of domains and boundaries within the spin-coated films. In this configuration the thiophene backbones are perpendicular to the interface and we are able to resolve the lattice formed by terminal alkyl groups. We argue that the combination of imaging PT with backbones parallel to the interface (on hBN) and perpendicular (thin

film) provides a coherent discussion of this important class of polymer and represents a significant new contribution. We hope the referee will re-consider their evaluation taking into account the results in Figs. 3-4 and pages 6-9. We have modified our title and abstract to highlight the inclusion of thin films in our investigations.

Reviewer #3 (Remarks to the Author):

The manuscript describes the use of dynamic AFM, working at higher eigenmodes of the cantilever, to image the molecular and sub-molecular organisation of polythiophenes in two- and three-dimensional films. Polythiophenes are of broad interest due to their extensive use in organic electronic devices and have been the target for several in-depth structural investigations with a variety of techniques. While scanning probe methods have been previously used to image molecular organisation in 2D films of these materials, this is (to my knowledge) the first report of molecular resolution in real space on a 3D polythiophene film. This is significant, as (as the authors point out) it paves the way for correlating structure with device performance in the future.

There are some areas where further discussion would be useful and interesting, specifically:

1) In figure 1a, the apparent length of the polymer chains is quite variable – from <10nm to >100nm. Is this in line with the molecular weight and polydispersity of the material used?

2) The features in figure 1b appear quite markedly different to those in figure 1f (no alternating features along the backbone in 1b, and no visible side chains in 1f). Is this difference due to the different imaging conditions (amplitude, eigenmode etc.) used for the different images? If so, some brief discussion of how the imaging conditions affect the features that are resolved should be included.

3) In the discussion of figure 1f, the rows of bright alternating features are assigned to sulphur atoms. While the spacing along the chain supports this interpretation, the spacing across the chain appears to be significantly larger in figure 1f than is suggested by the structural models in figures 1d and 2a. Some discussion of the possible origin of this discrepancy should be included.

It would also be useful to include some further information, specifically:

a) Colour scale values should be included in the figure captions for all AFM images, and a scale and units should be added to the section in figure 1f.

b) A section covering AFM should be included in the Methods, including typical parameters such as typical values of the amplitude setpoint, scan rate, scan size, number of pixels etc and details of how the amplitude sensitivity was determined for higher eigenmodes.

Overall, the data are of extremely high quality, the analysis is robust and well described and the manuscript is well written. I believe this paper will be of interest in the field, both in terms of the imaging methods employed and the structural information that is revealed.

We thank the referee for their positive comments. Our responses follow below:

1. The manufacturer's information specifies the molecular weight as being in the range 30000-100000 which would correspond to lengths of 100 – 300 nm. We do observe strands with lengths in this rather broad range but as the referee points out there are sections which are smaller, although it is not always straightforward to identify end points with confidence. At this stage we do not feel confident in making a detailed analysis of length distribution since our work has focused more on the imaging itself but this remains a longer term objective now that the necessary resolution has been demonstrated.
2. The images were taken using different protocols and we have included more details of the relevant parameters and cantilevers in additional sections introduced on pages 4 and 5. We have

now provided the exact imaging conditions in the relevant figure captions for all AFM scans. The highest resolution is acquired using low amplitude third harmonic excitation of the cantilever.

3. We have now added some discussion in the main text to explain this point; we argue that the apparent spacing across the chain arises from the interaction with lone pairs of electrons associated with the sulphur atom resulting in a convolution effect and the displacement of the maximum contrast point by approximately 0.1 nm from the expected position of the nuclear position of the atom.

On the additional points:

- a. We prefer to use profiles to convey height information and included these for the topographic images; this is our preference since the colour scale bars are difficult for readers to interpret quantitatively. We will also make the original images publicly available would prefer not to include colour scale values on AFM images. All the relevant height information is provided on profiles included in figures.
- b. Since the parameters vary from image to image we have included these values in the relevant figure captions; a brief discussion of the estimation of the amplitude sensitivity is now included in the text.